# Nanoscale Phase Separation and Lattice Complexity in VO₂: The Metal–Insulator Transition Investigated by XANES via Auger Electron Yield at the Vanadium L₂₃-Edge and Resonant Photoemission

**Augusto Marcelli** [1,2,3,*] ⓘ, **Marcello Coreno** [3], **Matus Stredansky** [4,5] ⓘ, **Wei Xu** [2,6] ⓘ, **Chongwen Zou** [7], **Lele Fan** [8], **Wangsheng Chu** [7], **Shiqiang Wei** [7], **Albano Cossaro** [5], **Alessandro Ricci** [2], **Antonio Bianconi** [2,9,10] ⓘ and **Alessandro D'Elia** [4,5]

[1]  Laboratori Nazionali di Frascati, Istituto Nazionale di Fisica Nucleare, 00044 Frascati, Italy
[2]  RICMASS, Rome International Center for Materials Science Superstripes, Via dei Sabelli 119A, 00185 Rome, Italy; xuw@mail.ihep.ac.cn (W.X.); phd.alessandro.ricci@gmail.com (A.R.); antonio.bianconi@ricmass.eu (A.B.)
[3]  ISM-CNR, Istituto Struttura della Materia, LD2 Unit, Basovizza Area Science Park, 34149 Trieste, Italy; marcello.coreno@cnr.it
[4]  Department of Physics, University of Trieste, Via A. Valerio 2, 34127 Trieste, Italy; matus.stredansky@studenti.units.it (M.S.); delia@iom.cnr.it (A.D.)
[5]  IOM-CNR, Laboratorio Nazionale TASC, Basovizza SS-14, km 163.5, 34012 Trieste, Italy; cossaro@iom.cnr.it
[6]  Beijing Synchrotron Radiation Facility, Institute of High Energy Physics, Beijing 100049, China
[7]  National Synchrotron Radiation Laboratory, University of Science and Technology of China, Hefei 230026, China; czou@ustc.edu.cn (C.Z.); chuws@ustc.edu.cn (W.C.); sqwei@ustc.edu.cn (S.W.)
[8]  Key Laboratory for Advanced Technology in Environmental Protection of Jiangsu Province, Yancheng Institute of Technology, Yancheng 224051, China; fanle@mail.ustc.edu.cn
[9]  IC-CNR, Istituto di Cristallografia, Via Salaria km 29, 00015 Roma, Italy
[10]  National Research Nuclear University Mephi, Kashirskoe shosse 31, 115409 Moscow, Russia
*   Correspondence: marcelli@lnf.infn.it; Tel.: +39-06-9403-2737

**Abstract:** Among transition metal oxides, VO₂ is a particularly interesting and challenging correlated electron material where an insulator to metal transition (MIT) occurs near room temperature. Here we investigate a 16 nm thick strained vanadium dioxide film, trying to clarify the dynamic behavior of the insulator/metal transition. We measured (resonant) photoemission below and above the MIT transition temperature, focusing on heating and cooling effects at the vanadium L₂₃-edge using X-ray Absorption Near-Edge Structure (XANES). The vanadium L₂₃-edges probe the transitions from the 2p core level to final unoccupied states with 3d orbital symmetry above the Fermi level. The dynamics of the 3d unoccupied states both at the L₃- and at the L₂-edge are in agreement with the hysteretic behavior of this thin film. In the first stage of the cooling, the 3d unoccupied states do not change while the transition in the insulating phase appears below 60 °C. Finally, Resonant Photoemission Spectra (ResPES) point out a shift of the Fermi level of ~0.75 eV, which can be correlated to the dynamics of the 3d// orbitals, the electron–electron correlation, and the stability of the metallic state.

**Keywords:** vanadium dioxide; resonant photoemission; metal–insulator transition; XANES; Auger electron yield; strained film; phase separation

## 1. Introduction

Transition metal (TM) oxides offer a wide spectrum of phase-separated systems with characteristic anomalies in different properties such as the electrical resistivity and/or the optical transmission. These phenomena originate from electronic correlation and interactions involving spin, lattice, and charge degrees of freedom. Spatially separated regions with distinct structural, magnetic, and electronic properties occur in these systems, which can be described as a multiscale phase separation between two (or more) phases that have a comparable free energy. This heterogeneity of the material may extend from the atomic scale to the mesoscale domain, indicating arrested phase separations, typical of TM oxides where complex textures emerge from the coexisting phases. The presence of a network of multiple domains plays a role in the dynamics of the phase transformations, and it is the origin of the strong anomalies in the transport, magnetic, and structural properties, which are characteristic of many complex systems.

Among TM oxides, vanadium dioxide ($VO_2$) is the typical correlated electronic material, where a metal to insulator transition (MIT) occurs in its bulk phase near room temperature [1–4]. Around 67–68 °C, the vanadium dioxide undergoes the electronic metal–insulator transition upon heating or cooling with hysteretic behavior and a change in the electrical conductivity by several orders of magnitude, coupled to a Structural Phase Transition (SPT) from the monoclinic to the rutile phase. Since the discovery of the MIT transition more than 50 years ago by Morin and Westman [1,2], $VO_2$ has attracted a lot of interest because of its strong electron correlation; recently the interest has increased because of the wide number of different possible applications in optics, as detectors or sensors, and in novel memory devices based on the occurrence of the reversible MIT transition [5–11]. However, since its discovery, and even now, the nature of this electronic/structural transition remains an open question. Based on the available experimental results, the driving mechanism of the MIT transition in $VO_2$ has been considered an electron-correlation-driven Mott transition [3,4] or a structural distortion-driven Peierls transition, or a cooperation of both mechanisms. Actually, the scenario can be even more complex if we consider films of $VO_2$. As pointed out by Fan et al. [8], investigating thin and ultrathin $VO_2$ films grown on oriented $TiO_2$ substrates, the strain dynamics also play a role and, in these films, the MIT process is modulated continuously via the interfacial strain [9,12,13]. The presence and role of different phases and the relationship between the phase transition temperature and strain have been investigated during film growth. Moreover, the interfacial strain strongly affects the electronic orbital occupancy, which changes also the electron–electron correlation and controls the phase transition temperature. Recently Mengmeng Yang et al. [9] investigated the MIT mechanism in a 13 nm thick strained $VO_2$ film on $TiO_2$. With temperature-dependent synchrotron radiation high-resolution X-ray diffraction data and Raman spectroscopy, the authors suggested that the structural phase transition in the temperature range near the MIT is suppressed by epitaxial strain, and the electronic transition triggers the MIT in strained films [9].

In this paper we will describe the existing scenario and will attempt to clarify the dynamic behavior, investigating the insulator/metal transition in a $VO_2$ thin film combining X-ray absorption spectroscopy (XAS) and (resonant) photoemission experiments at different temperatures. In particular, we will show Resonant Photoemission Spectroscopy (ResPES) and high-resolution XANES (X-ray Absorption Near-Edge Structure) spectroscopy at the V $L_{23}$- and O K-edges vs. temperature, i.e., below and above the temperature of the MIT transition while heating and cooling the film. The ResPES across the V 2p–3d threshold and, in particular, the valence band and the core level V 2p spectra will help to provide a consistent picture of the filled density of states (DOS) and of other electronic parameters of vanadium oxide films [12]. At variance, the XANES technique has been chosen because of its unique ability to probe simultaneously both electronic and structural properties, with the elemental selectivity that, as we will see in the forthcoming, is particularly important for monitoring this complex electronic and structural process. Actually, as mentioned above, the $VO_2$ may simultaneously undergo an MIT and an SPT from a monoclinic insulator to a metallic rutile structure going below and above the MIT temperature.

## 2. Results

The sample investigated is a thin film of $VO_2$ deposited on a $TiO_2$ (001) substrate (see Section 4). Among the many possible substrates, the rutile phase of $TiO_2$ is probably the best on which to grow $VO_2$ films because of its stable thermal properties and similar lattice parameters. The latter are key parameters to control the interfacial strain/stress of an epitaxial film. Many studies report the growth of high quality $VO_2$ films on $TiO_2$ substrates, and the phase transition temperature depends on the involved interfacial strain/stress. As an example, Muraoka et al. reported the growth of a $VO_2$ film on a $TiO_2$ (001) surface, observing a decrease of the phase transition temperature down to room temperature [13].

We report here the investigation of a 16 nm film grown on $TiO_2$ substrate and, as deeply discussed in [8] for films of similar thickness from 1.6 nm to 74 nm grown on an oriented $TiO_2$ substrate, the role of the interfacial strain is important. Actually, in ultrathin films, i.e., thinner than 10 nm, a fully strained behavior is observed with an MIT temperature near room temperature, while thicker films, i.e., greater than 24 nm, are characterized by diffraction patterns that point out an almost fully relaxed interface, higher MIT transition temperatures, and lower resistance in the metallic state [6].

One possibility to extract detailed electronic–structure information from a correlated 3d electron system is given by the resonant photoemission spectroscopy. This technique takes advantage of the fact that an important decay channel in the photoemission process is associated with an Auger-like matrix element in which the core hole is refilled by an electron from the final-state shell (3d) and another electron is ejected into the continuum. The ResPES spectra of our $VO_2$ film across the V 2p–3d threshold and normalized to the incoming photon flux are shown in Figure 1. They have been collected using the measurement mode called "constant initial state" (CIS) [14,15]. As can be seen in the Figure 1, where a series of photoemission spectra are taken with increasing photon energy, as the photon energy $h\nu$ goes through a cross section resonance, the emission from the resonating state is modulated. In this case, the comparison of photoemission cross sections collected at different excitation energies clearly shows the enhancement of the V 3d electronic structure going from the insulating (30 °C) to the metallic state (90 °C). In fact, the V 3d photoemission cross section strongly resonates at ~2 eV due to the interference between the direct excitation, given by

$$3p^6 3d^n + h\nu \rightarrow 3p^6 3d^{n-1} + e \quad (1)$$

and the photo-absorption channel, which describes the emission of a 3d electron from the vanadium atom:

$$3p^6 3d^n + h\nu \rightarrow 3p^5 3d^{n+1} \rightarrow 3p^6 3d^{n-1} + e. \quad (2)$$

The ResPES technique is then particularly suitable for investigating the behavior of correlated systems such as vanadium oxides [12] and other TM oxides [14]. A description of this resonance effect is provided by the Fano theory of photoabsorption. As we will show in the next section, if a series of photoemission spectra are taken, increasing the photon energies, then as $h\nu$ goes through the cross section resonance, the emission from the resonating state is modulated. With the CIS acquisition procedure, the photon energy and the detected electron kinetic energy ($E_{KE}$) are simultaneously varied so as to keep the ionization energy ($E_I$) constant:

$$E_I = h\nu - E_{KE}. \quad (3)$$

In analogy to the discussion of Eguchi et al. in [16]—about ResPES spectra of a 10 nm thick $VO_2$ film, epitaxially grown on the (001) surface of a $TiO_2$ single-crystal substrate—in Figure 1, we compare selected photoemission spectra of this film in both metallic and insulating phases. They correspond to specific photon energies in the XAS profile at the $L_{23}$-edges of vanadium (see the Figure 1c).

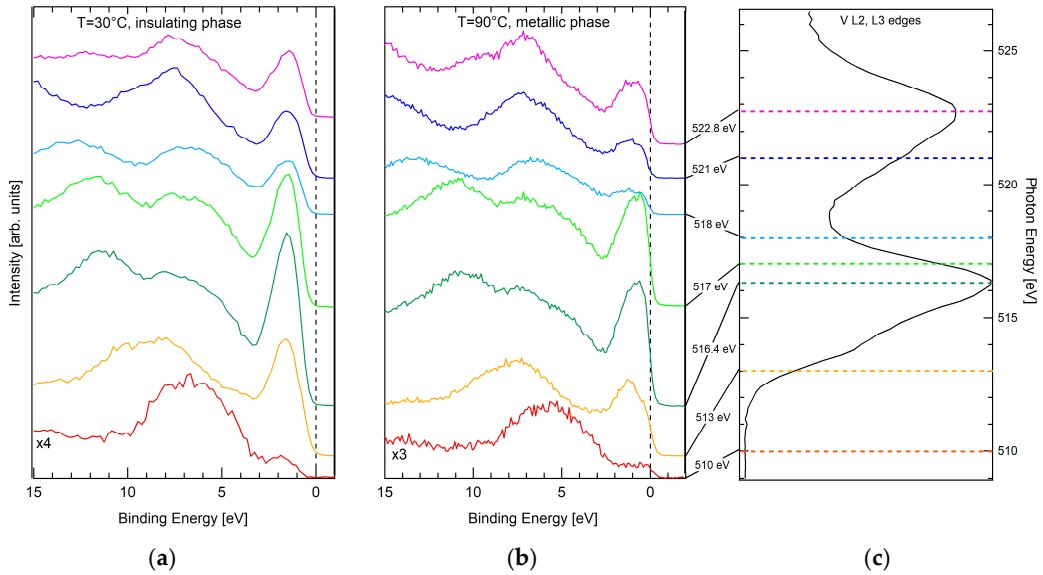

**Figure 1.** Resonant Photoemission Spectroscopy (ResPES) spectra of the 16 nm $VO_2$ film in the binding energy range (15; 1 eV). Spectra have been collected on resonance to the energy of the V $L_3$-edge (516.4 eV) and $L_2$-edge (522.8 eV) and off resonance (521, 518, 517, 513, and 510 eV) both in the insulating (*T* = 30 °C, (**a**)) and in the metallic phase (*T* = 90 °C, (**b**)). In the panel (**c**) is shown the XANES spectrum of the insulating phase (Auger yield at 464 eV). The spectra collected at 510 eV have been magnified (x4 insulating phase and x3 metallic phase). All spectra have been normalized to the incident photon flux.

While the off-resonance spectrum is very similar to the standard PES spectrum of the bulk $VO_2$ [17], the other spectra show a resonance enhancement of the V 3d feature (in the range 0–2 eV) as a function of *hv* with a maximum around ~516 eV ($L_3$ peak in XAS). This behavior confirms the 3d electron character of this feature. At higher *hv*, a weak Auger feature shows up at higher binding energy, as outlined by black arrows in the ResPES map collected at 30 °C in Figure 2. To the best of our knowledge, ResPES spectra of similar $VO_2$ films are available also in [17], while the ResPES map in Figure 2 is the first published for a thin $VO_2$ film. From the analysis of these photoemission spectra, it is possible to evaluate the gap associated with the MIT process in our film, which is 0.75 ± 0.25 eV; this is in agreement with the value of 0.6 eV measured in a single crystal of $VO_2$ in [17], where Koethe et al. already pointed out this anomalously large energy value compared with the energy scale of the MIT temperature (~30 meV).

The photoemission experiments were performed to characterize the film in terms of density of states, electronic configurations, and electron correlation while going from a low-*T* monoclinic insulating phase to a high-*T* rutile-like structure. Several direct-photoemission (PES) experiments have been reported in the literature for $VO_2$ [17,18] and other vanadium oxides [12], but none of them are yet satisfactorily described by a calculated density of states or a model. As a consequence, in addition to an accurate ResPES experiment, we also performed XANES experiments to probe the local and partial empty density of states of vanadium and oxygen atoms [19].

The XAS experiments were performed using the Auger yield acquisition technique, collecting the vanadium Auger electrons with the kinetic energy of 464 eV and the oxygen Auger electrons with the kinetic energy of 507 eV. As will be discussed in the next section, this choice guarantees to the spectra an improved chemical and surface sensitivity with respect to any other method. For this film, the XANES spectra were different. The one acquired using the V Auger yield (see the Figure 3a) is similar to the spectra of $VO_2$ published in the literature [20]. At variance, since the O K-edge (~530 eV) is close in energy to the V $L_2$-edge, the O K-edge (see Figure 3b) is superimposed on the end of the

spectrum of the V L-edges (~512–527 eV) and, without doubt, the tail of the latter affects both the shape and the intensity of the oxygen K-edge spectrum.

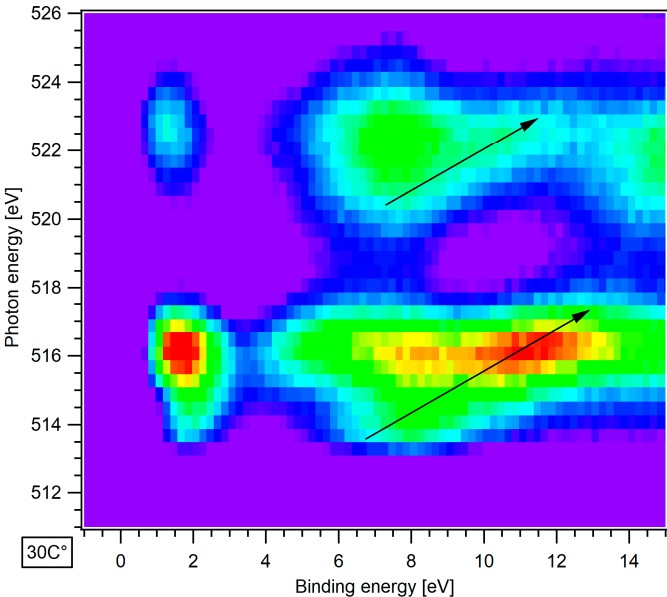

**Figure 2.** ResPES map of the 16 nm VO$_2$ film collected in the insulating phase (30 °C) in the energy range from 509 eV to 526 eV and in the binding energy range (−1; 15 eV). The black arrows outline the dispersions of the Auger lines of the incoherent V LVV Auger emissions, i.e., the L$_3$M$_{45}$M$_{45}$ and the L$_2$M$_{45}$M$_{45}$.

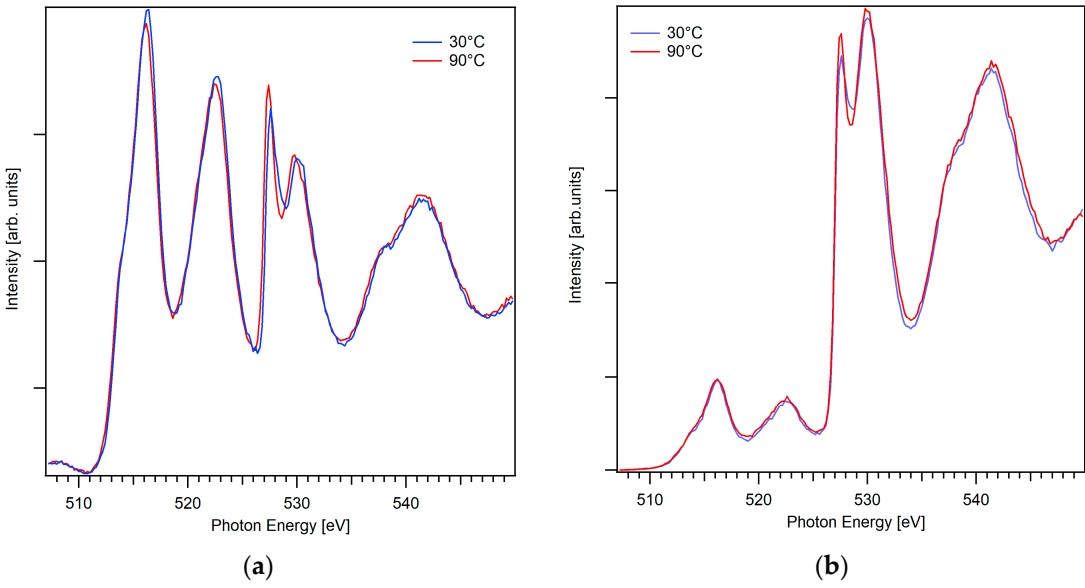

**Figure 3.** Comparison of XANES spectra of the 16 nm thick film collected using the Auger electron yield: (**a**) vanadium Auger yield at 464 eV (**b**) and oxygen Auger yield at 507 eV both collected at 30 °C and 90 °C.

The spectra collected with the V Auger electrons with the kinetic energy of 464 eV are thus suitable for investigating only vanadium edges. At variance, the XANES collected using the O Auger yield are characterized by an intense signal from the O K-edge and a much lower contribution from the vanadium L-edges. This is the ideal situation to collect O K-edge XANES spectra. To the best of our knowledge, this is the first time that Auger spectra have been compared for vanadium oxide.

We demonstrate here that the optimized way to obtain information at the oxygen K-edge is to collect spectra using the oxygen Auger yield at 507 eV. The scenario also holds true in comparison with experiments performed in transmission, as in [21], where a film of 40 nm of $VO_2$ obtained by oxidizing a thin film of vanadium on a silicon nitride membrane has been measured. As shown in Figure S8 in [21], the cross section of the O K-edge is much smaller than the V L-edge and is affected by changes at the V edges occurring during the transition. As shown in Figure 3, spectra collected with the V Auger electrons with the kinetic energy of 464 eV are representative of XANES features up to the energy of the O K-edge (~526–527 eV) for this film of 16 nm. Clearly, spectra collected with the two Auger yields are different.

　　　To investigate the MIT transition by looking at the empty 3d local density of states of vanadium, we compared the behavior of the $L_{23}$-edge vanadium absorption spectra. Indeed, as discussed above, the better way to obtain information is to collect spectra using the V Auger yield. In the two panels of Figure 4, we compare the V $L_{23}$-edge absorption spectra collected during the heating (Figure 4a) and cooling (Figure 4b) procedures from the temperature of 30 °C (taken as the reference) up to 90 °C. As expected for a system that exhibits hysteretic behavior, the spectra collected with the two procedures do not show a specular behavior. Major differences seem to occur during the cooling process. Actually, looking at the hysteresis curves of similar films, this result is not unexpected [8]. The 16 nm thick film (Figure 3a in [6]) exhibits an MIT transition just above 30 °C, and the transition to the metallic state is almost completed at ~70 °C. While cooling, the transition starts around ~60 °C and is almost completed at ~30 °C.

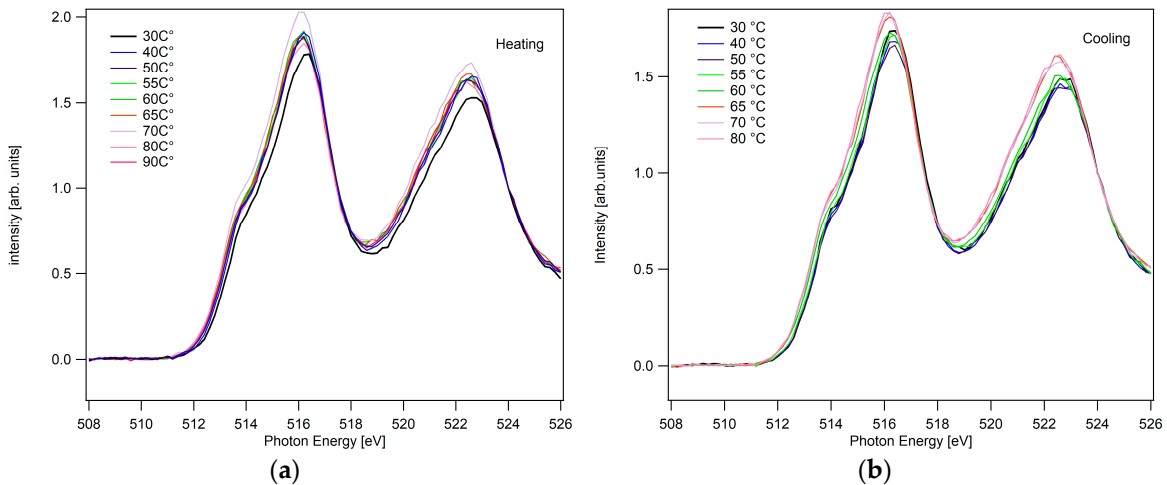

**Figure 4.** Comparison of XANES spectra of the 16 nm thick film collected using the Auger electron yield at 464 eV: vanadium Auger yield spectra during the heating process up to 90 °C (**a**) and during the cooling procedure down to 30 °C (**b**).

## 2.1. Experimental Methods

### 2.1.1. Resonant Photoemission Spectroscopy (ResPES)

　　　In a ResPES experiment, the photon energy is tuned through a core-level excitation, i.e., through an X-ray absorption edge, continuously changing the photon energy in a wide energy range [22]. The coherent resonance of the two final states, summarized in Equations (1) and (2), leads to a characteristic variation of the photoemission intensity with the photon energy, known as the Fano line-shape. The behavior can be used for the assignment of valence band states to individual components of the solid or particular final states. For a given configuration $3d^n$ of the valence shell, the ResPES effect can be described as an interference of the direct PE process (Equation (1)) and the second excitation channel (Equation (2)), which leads to the same final state when the photon energy hν is swept through the absorption threshold of a core level c. The first step of the excitation

produces an intermediate state, which decays in the second step through a Coster–Kronig (CK) or super-Coster–Kronig (SCK) process. Coherent superposition of channels in Equations (1) and (2) can lead to a resonant enhancement of the $3d^{n-1}$ final state, especially if the cross section for the absorption process is strong ("giant resonance") [14]. Due to the local character of the ResPES process, it is possible to distinguish TM 3d-like states in the valence band from those having a ligand character. In the case of $VO_2$, each vanadium ion is nominally $V^{4+}$ in the $3d^1$ configuration—the main contribution to the ground-state wavefunction. ResPES further allows one to separate the different final states with a hole contribution on the ligand orbitals (L̲), although a strong mixing among different L̲ configurations already exists in the ground state of vanadium dioxides. Previous ResPES experiments on vanadium oxides performed at the V 3p threshold already pointed out a strong V 3d admixture with the O 2p part of the valence band spectra [12]. Moreover, because the cross section of the 2p photo-absorption edge in TMs is more intense compared with the 3p edge, a large resonant enhancement in the PES spectra occurs near the 2p threshold, as shown in Figure 1. The PES spectra in Figure 1 and those contained in the map in Figure 2 have been collected in the CIS mode within three different intervals: from 509 eV to 512 eV (step 1 eV), from 512.4 eV to 535 eV (step 0.4 eV), and from 535.8 eV to 550 eV (step 0.8 eV), changing the kinetic energy from 449 eV to 509 eV (step 0.1 eV). Measurements have been performed at the ANCHOR end-station of the ALOISA beamline [23] at the Elettra synchrotron radiation facility. Electrons were collected at normal emission with the photon beam linearly polarized in the scattering plane and impinging the sample at the magic angle (35°). A PSP Vacuum 120 mm electron analyzer with a 2D delay line detector was used. Measurements were performed at constant pass energy ($E_p$ = 20 eV) with an overall resolution of 0.25 eV.

### 2.1.2. Auger Electron Yield X-ray Absorption Spectroscopy

The first surface XAS experiment with high surface sensitivity was carried out in 1977 at SSRL by Bianconi et al. through detection of the Auger electron yield (AEY) [24]. In this study, the absorption spectrum of the Al surface atoms in the top monolayers of an Al crystal was distinguished from the Al bulk spectrum. Indeed, the energy of the Auger electrons is characteristic of a particular atom, and the theoretical relation between Auger electron yield and surface absorption coefficient of the photoabsorber was independently predicted by Lee [25] and Landman [26]. Both the total electron yield (TEY) and the AEY methods were used to measure surface X-ray absorption spectra of different atomic species chemisorbed on solids. Actually, in the soft X-ray energy range, e.g., <4000 eV, the Auger recombination has a higher probability than the radiative recombination, and the detection of elastically emitted Auger electrons is an efficient way to measure the surface absorption coefficient. The Auger line is selected by an electron-energy analyzer operated in the constant final state (CFS) mode with an energy window of a few eV. Although the AEY technique has the smallest signal rate, it offers the largest S/N ratio among all electron-yield techniques. This choice also guarantees improved chemical and surface sensitivity.

### 3. Discussion

X-ray absorption spectroscopy has proven to be a great tool for studying unoccupied conduction bands. XANES can be particularly useful for probing and mapping the changes in the electronic structure of an MIT transition, providing a correlation between macroscopic observables and microscopic models associated with coexisting metastable configurations, which are tuned by the thickness-dependent misfit strain and stress distributions induced by the mismatch between thin film and substrate [26–28]. Since, in the $VO_2$ electronic MIT-driven transition, a relatively large change in the electronic configuration between the metallic and insulating phases is expected, an accurate measurement of the difference among V 2p X-ray absorption spectra will probe changes of the local 3d orbital occupancy. Yet there are limited reliable XAS data for thin $VO_2$ films, whose MIT characteristics vary significantly from those of bulk crystals and with respect to different preparations and substrates.

A recent investigation using X-ray absorption spectro-microscopy [21] showed that it is possible to induce an electronic transition in a thin film of $VO_2$ during a heating–cooling cycle without inducing a structural transition. Also, this experiment demonstrates the relevance of the XANES technique, characterized by high spectral, time, and spatial resolution, for probing phase-separated materials or materials with complex multidimensional phase diagrams [29]. Another interesting investigation providing data on the $VO_2$ absorption edges tried to identify differences due to the film preparation conditions and, as in our study, to probe the evolution of XANES spectra vs. temperature, looking for irreversible changes occurring after multiple thermal cycles [30].

The XAS spectra in Figure 4 describe the V $L_{23}$-edges, i.e., the transitions from 2p to 3d levels characterized by two pronounced maxima at ~516 and ~523 eV, which roughly correspond to the electron excitations from spin–orbit split levels $2p_{3/2}$ and $2p_{1/2}$, respectively. The final states of these transitions are the *d*-projected empty density of states (DOS) of the valence levels, perturbed by the core hole created in the V 2p core level. In Figure 5, we show the map of the differences among the XAS spectra in Figure 4, collected in both the heating and cooling processes. The image has been obtained by plotting the differences among each spectrum vs. the first spectrum collected at 30 °C taken as a reference, e.g., (XAS@30 °C–XAS@40 °C), (XAS@30 °C–XAS@50 °C), etc. Looking at the dynamics of the 3d empty DOS, it is evident the decrease of the empty density of states both at the $L_3$ and at the $L_2$ is already starting at 40 °C; this is as expected for a transition to a metallic state, which occurs at a lower temperature for a partially strained film. The decrease is continuous up to a maximum around 70 °C, although it does end only when the cooling procedure starts. As expected due to the hysteretic behavior of this thin film, in the first stage of the cooling, the empty DOS does not change, and the transition in the insulating phase smoothly appears at temperatures below 60 °C.

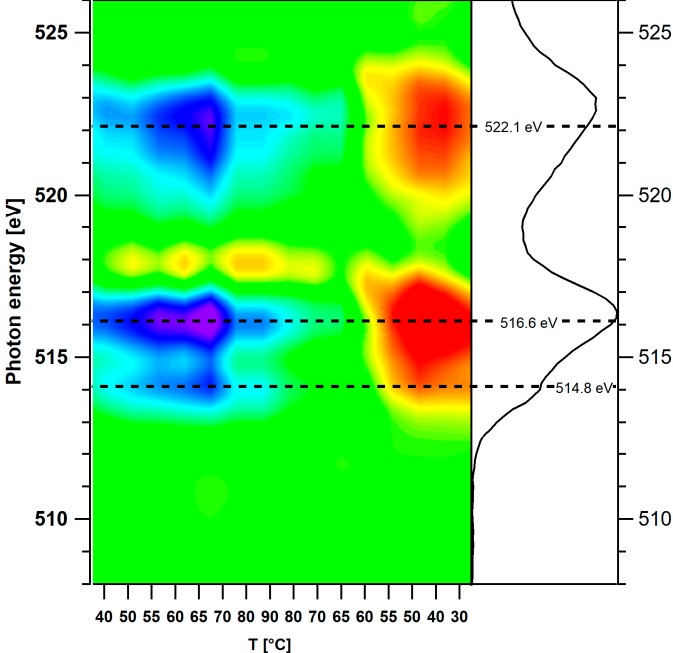

**Figure 5.** Map of the differences of the V $L_{23}$ XANES spectra of the $VO_2$ 16 nm film collected using the vanadium Auger yield at 464 eV. From top to bottom, the evolution of the difference among XANES spectra in the heating and cooling processes. Differences are obtained by taking the first spectrum collected at 30 °C as the reference (see text for more details).

The signature of the MIT transition at the V $L_{23}$-edge, enhanced by differences detected in the map in Figure 5, is accompanied by the results obtained by the ResPES spectra of this film; together, these point out a shift of the Fermi level of ~0.75 eV, which can be correlated to the shift of the $d_{//}$ band and its increase of occupancy. The $d_{//}$ band is very sensitive to the variation of the c axis of

the Rutile phase of the $VO_2$. Its overlap with the $\pi^*$ band increases the number of itinerant electrons, so that the electron–electron correlation decreases and the metal state stabilizes [31].

Looking at the metallic states in [32], it is shown that the MIT change in the resistance of thick films of $VO_2$ is similar to the change measured in thin films. Indeed, the change in the resistance of our ~16 nm thick film is <1 k$\Omega$ [8], while in the work of Ruzmetov et al. [32], thicker films (~100 nm) show a change in resistance at the MIT comparable with that of a 74 nm film. Actually, thick $VO_2$ films exhibit less strain, and the role of the oxygen deficiency cannot be simply evaluated. Indeed, looking at Figure 1 of [32], we may see that the resistance in the insulating phase goes from <1 k$\Omega$ to >1000 k$\Omega$. Moreover, the two optimized samples exhibit a difference in the resistance of two orders of magnitude.

Finally, we would like to remark that the spatial inhomogeneous lattice distribution [33] showing up at the MIT in strongly correlated metals is expected in the frame of arrested nanoscale phase separation in a multiband Hubbard model for correlated charge carriers [31,33]. Moreover, the arrested electronic and structural phase separation is predicted to show up [34,35] in a critical range of strain and doping around the Lifshitz transition. Our results confirm that the MIT in $VO_2$ is associated with the electronic topological Lifshitz transition. In fact, at the MIT, the upper 3d band crosses the Fermi level and a new small vanadium 3d Fermi surface appears, giving the Lifshitz transition.

## 4. Materials and Methods

The $VO_2$ film we investigated had a thickness of 16 nm and was deposited on a clean substrate of $TiO_2$ (001) by rf-plasma-assisted oxide Molecular Beam Epitaxy (MBE) (assembled in house for oxide materials growth, at University of Science and Technology of China, Hefei, China), working at the base pressure <3 $\times$ 10$^{-9}$ Torr. At a constant growth rate, the thickness was controlled by adjusting the deposition time in a range from several unit cells to tens of nanometers. The interfacial cross section was investigated with a high-resolution scanning transmission electron microscope (STEM) (JEOL Ltd., Tokyo, Japan). High-angle annular dark-field (HAADF) scanning transmission electron microscopy (STEM) images were taken on a JEM ARM200F (JEOL Ltd., Tokyo, Japan) with a probe aberration corrector, while the diffraction pattern was acquired on a JEM 2100 TEM (JEOL Ltd., Tokyo, Japan). More details of the epitaxial film preparation, performed at the University of Science and Technology (Hefei, China), are reported elsewhere [36].

**Acknowledgments:** This research has been performed within the Elettra Proposal #20160373 "Dynamic competition between insulating and metallic phase in $VO_2$" at the ALOISA beamline. W.X. acknowledges the NSFC grant No. U1532128 and INFN for the financial support.

**Author Contributions:** A.M., M.C., S.W. and A.R. conceived and designed the experiments; A.M., M.C., M.S., A.C. and A.D. performed the experiments; M.S., W.X., A.R. and A.D. analyzed the data; C.Z., L.F. and W.C. synthesized the films; A.M. and A.B. wrote the paper. All authors have read and approved the final manuscript.

**Conflicts of Interest:** The authors declare no conflict of interest.

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
