# Peer review of "Nanoscale Phase Separation and Lattice Complexity in VO2: The Metal–Insulator Transition Investigated by XANES via Auger Electron Yield at the Vanadium L23-Edge and Resonant Photoemission"

_condensedmatter, doi:10.3390/condmat2040038_

Round 1

Reviewer 1 Report

The authors have presented a photoemission study of VO2 strained films. This is a material seeing lots of attention in the CMP field due to the phase transition physics of current interest. the study here will contribute to further experimental data on spectroscopic properties and so is useful in that regard. I have some comments for the authors to consider:

- Prior XAS work by Ruzmetov et al on VO2 (Physical Review B77(19), p.195442) should be cited and discussed in the paper. They have clearly shown how the band structure is different for different V-O ratios and it would be important to compare this to the strain effects presented in this study, since in both cases, the metallic state resistance is strongly affected. So, it will be usefu to know if the resistance changes seen in strained films is similar or different to non-stoichiometry effects.

- the -0.75 eV shift reported here: is that change in band gap across IMT? it seems a bit higher than typical gaps reported for VO2. it will also be useful for the authors to compare this to work function changes across IMT reported for VO2 to see if they can map the electronic structure better. It will also be useful to know if the band edges move symmetrically across IMT from this study (if possible to estimate this).

Author Response

The reply is contained in the pdf in attachment.

Reviewer 2 Report

The paper by Marcelli et. al. studies the electronic structure of VO2 – a prototype correlated oxide – as it goes through the metal insulator transition (MIT) using X-ray absorption near edge structure (XANES). There is great interest in using the MIT of VO2 for applications in optics and memories, although the exact mechanism of its MIT is still under huge debate. The current results provide an interesting experimental probe into this issue, which makes it suitable for Condensed Matter. However, there are several major comments that should be addressed before I could recommend its publication.

First, as the authors discussed very well in the study, there is phase coexistence when VO2 goes through MIT. As a result, the XANES spectra would be an average of two different phases depending on the exact microstructure. How large is the X-ray spot size or footprint? Is there any spatial inhomogeneity in the XANES? Do the authors know if there is phase coexistence in their films?

It is unclear how the bandgap energy is extrapolated from Fig. 1.

While the authors discussed SPT vs MIT in the introduction, I don’t see any related data or discussion in the main text.

The reference is insufficient. The following paper should be cited when the authors discuss VO2’s application in optics, as detector, sensor, or novel memory devices: Proceedings of the IEEE 103, 1289 (2015) and Nature Communications 6, 8636 (2015).

There are many grammar mistakes and typos in the paper, which makes it difficult to follow. I suggest that the authors have the paper professionally edited. Here I just list a few examples:

“VO2 may undergo simultaneously to a MIT and to a SPT from a monoclinic insulator to a metallic rutile structure going below and above the MIT temperature” should be “VO2 may undergo simultaneously though a MIT and a SPT from a monoclinic insulator to a metallic rutile structure going below and above the MIT temperature”

As deeply discussed in Ref. [8]” should just be “As deeply in Ref. [8]

“thinner then 10 nm” should be “than”

Author Response

The reply is contained in the pdf in attachment.

Comment by the editorial office: Please note that we have included also the replies to the second reviewer since the authors are referring to it while replying to your report.

Round 2

Reviewer 1 Report

The authors have provided a response to my comments. I agree the thicker films will show less effect of strain driven modification of IMT. It is a pity they cannot make any additional conclusions from these detailed studies on XANSES concerning the band edge changes across IMT.

Nevertheless, I have no further comments on the manuscript.

Reviewer 2 Report

My comments have been addressed and I recommend its publication.